# Identification of system-level features in HIV migration within a host

**Ravi Goyal** [1]*, **Victor De Gruttola**[2], **Sara Gianella**[1], **Gemma Caballero**[1], **Magali Porrachia**[1], **Caroline Ignacio**[1], **Brendon Woodworth**[1], **Davey M. Smith**[1], **Antoine Chaillon**[1]

**1** Division of Infectious Diseases and Global Public Health, University of California San Diego, La Jolla, CA, United States of America, **2** Herbert Wertheim SPH and Human Longevity Science, University of California San Diego, La Jolla, CA, United States of America

* r1goyal@health.ucsd.edu

## Abstract

### Objective

Identify system-level features in HIV migration within a host across body tissues. Evaluate heterogeneity in the presence and magnitude of these features across hosts.

### Method

Using HIV DNA deep sequencing data generated across multiple tissues from 8 people with HIV, we represent the complex dependencies of HIV migration among tissues as a network and model these networks using the family of exponential random graph models (ERGMs). ERGMs allow for the statistical assessment of whether network features occur more (or less) frequently in viral migration than might be expected by chance. The analysis investigates five potential features of the viral migration network: (1) bi-directional flow between tissues; (2) preferential migration among tissues in the same biological system; (3) heterogeneity in the level of viral migration related to HIV reservoir size; (4) hierarchical structure of migration; and (5) cyclical migration among several tissues. We calculate the Cohran's Q statistic to assess heterogeneity in the magnitude of the presence of these features across hosts. The analysis adjusts for missing data on body tissues.

### Results

We observe strong evidence for bi-directional flow between tissues; migration among tissues in the same biological system; and hierarchical structure of the viral migration network. This analysis shows no evidence for differential level of viral migration with respect to the HIV reservoir size of a tissue. There is evidence that cyclical migration among three tissues occurs less frequent than expected given the amount of viral migration. The analysis also provides evidence for heterogeneity in the magnitude that these features are present across hosts. Adjusting for missing tissue data identifies system-level features within a host as well as heterogeneity in the presence of these features across hosts that are not detected when the analysis only considers the observed data.

**Data Availability Statement:** The sequence data have been uploaded on Dryad at DOI: 10.5061/dryad.dncjsxm44. The code has been uploaded to Github at: https://github.com/ravigoyalgit/VMN.

**Funding:** This research is supported by grants from the National Institutes of Health (P01 AI131385, R01 AI147441, DP2 DA051915, R01 DK131532, P01 AI169609, P01 AI169609, P30 AI036214, UM1 AI164570, R01 AI147821, UM1 AI164559, R01 DA055491), James B. Pendleton Charitable Trust, and Department of Veterans Affairs.

**Competing interests:** The authors have declared that no competing interests exist.

## Discussion

Identification of common features in viral migration may increase the efficiency of HIV cure efforts as it enables targeting specific processes.

## 1 Introduction

Despite modern antiretroviral therapy (ART), HIV persists in deep tissue and cellular reservoirs. To cure HIV, we need to better understand reservoir persistence and its dynamics. Without ART, cell free virions can be detected in the bloodstream and circulate freely across the body. In addition, when ART is interrupted, HIV can rebound from deep tissues and repopulate other cell and tissue reservoirs [1, 2]. The biological mechanisms governing such replenishment and reseeding remain unclear. Identification of mechanisms driving viral migration would increase the efficiency of HIV cure efforts as it enables targeting specific processes. We take a system-level perspective based on statistical network science techniques applied to HIV genomic data to gain insight into such mechanisms by identifying common features of HIV migration. Specifically, our analysis investigates whether the frequency of system-level features associated with viral migration among the tissues occurs more or less often than would be expected by chance; such features include bi-directional flow between tissues, cyclical migration among several tissues, and hierarchical structure of migration. A challenge that arises in the identification of system-level features is the limited access to hard-to-reach tissues, which results in incomplete observation of viral migration events within hosts. Our analysis compares estimates associated with the frequency of viral migration feature with, and without, adjusting for missing data to investigate the potential impact of such adjustment. The analysis also uses an approach from meta-analysis to evaluate heterogeneity in frequency of these features across the hosts.

To investigate features associated with viral migration, we model the complex dependencies of HIV migration among tissues as a network. We refer to such networks as viral migration networks (VMNs). The use of network analysis has two important advantages over traditional statistical methods. First, network analysis enables statistical investigation of system-level features of replenishment and reseeding of HIV in tissues. Specifically, network analysis enables formal statistical testing of whether features observed in viral migration events occur at greater (or less) frequency than expected under a null hypothesis. For example, phylodynamic analysis for HIV sequencing data using Bayesian discrete diffusion models by [1] showed bi-directional migration within the central nervous system between the occipital lobe and frontal lobe. Network analysis allows an assessment of whether these observations would be expected by chance given the amount of migration among the tissues within a host [3, 4]. Second, network science methods are required to adjust for missing tissue data as standard approaches are not able to handle the complex dependencies among the tissues [5–8]. Ignoring missing data when complex dependencies exist by conducting complete case analysis has been shown to result in significant biases in the estimation of network features [9–13]. In this manuscript, we demonstrate that adjusting for missing tissue data can aid in (and potentially be necessary for) the identification of system-level features within a host as well as in assessing heterogeneity in these features across the hosts. In particular, the analysis presented below shows that observing evidence for the presence of several features depends on doing the adjustment.

A VMN for a participant is a directed network, wherein the tissues and blood are represented as nodes in the network, and the migration events are represented as directed edges from the tissue of egress to the tissue being reseeded or to blood. Migration events can be

inferred from viral genomic data using Bayesian discrete trait analyses (DTA), which estimate pairwise migration events between tissues [1]. We use of this approach on HIV full length envelope single genome sequencing data generated from blood and tissue samples from participants enrolled in the Last Gift (LG) cohort [14]. Our analysis that investigates features of the VMNs uses a common family of statistical network models called the exponential random graph models (ERGMs) [3, 4, 15, 16]. Only recently have there been studies using models to investigate an ensemble of networks; [17–19] these studies have generally focused on binary and complete networks. To our knowledge, this is the first study that makes use of network models to investigate an ensemble of value networks in the presence of missing data.

## 2 Results

### 2.1 Descriptive analyses of viral migration networks

The VMNs based on DTA applied to the LG cohort are shown in Fig 1. The nodes and edges represent tissues and migration events among the tissues, respectively. The node color indicates the biological system to which the tissue belongs (for example, central nervous system [CNS] or gut), the node size is proportion to the number of sequences within the sample, and the edge width denotes the number of migrations between the two tissues connected. As illustrated in the figure, the number of unsampled/missing tissues varies between 8 (LG04) and 26 (LG12).

The subsequent sections present results from the investigation of five features: (1) bi-directional flow between tissues; (2) preferential migration among tissues in the same biological system; (3) heterogeneity in level of viral migration related to HIV reservoir size; (4) hierarchical structure of migration; and (5) cyclical migration among several tissues. The results include an assessment of the heterogeneity in the presence and magnitude across individuals for each of these features as well as of the impact of missing data on the estimates associated with the features.

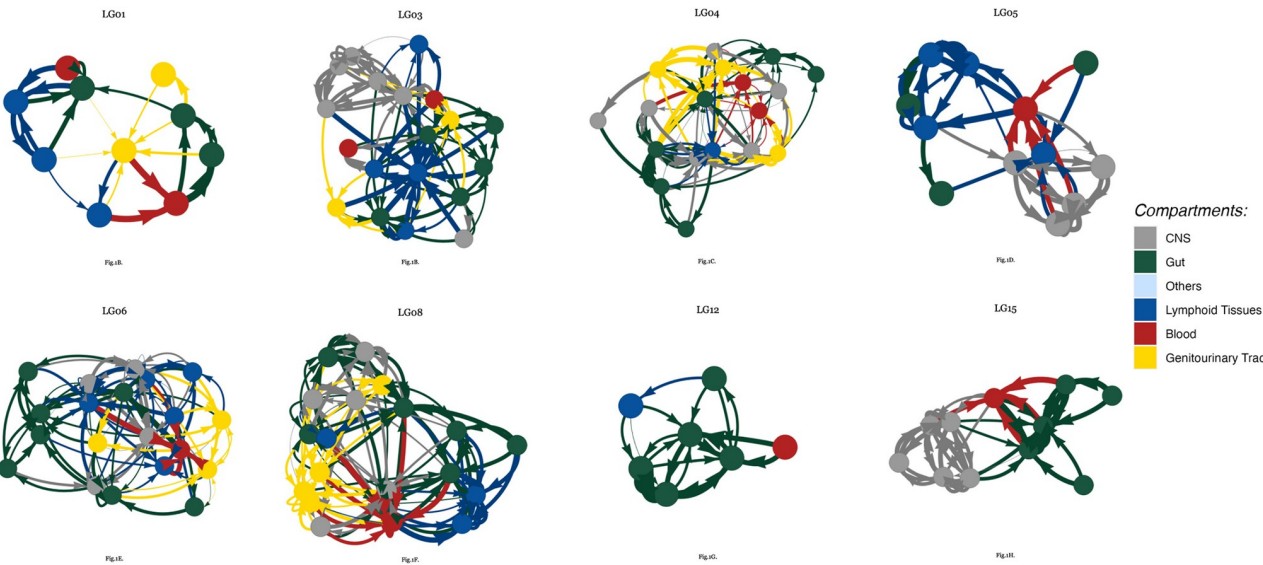

**Fig 1. Visual representation of the VMNs for each of the Last Gift participants.** The nodes and edges represent individual tissues and migration events among the tissues, respectively. The node color indicates the biological system to which the tissue belongs, the node size is proportion to the number of sequences within the sample, and the edge width denotes the number of migrations between the two tissues.

## 2.2 Viral migration is bi-directional (Reciprocity)

Our analysis provides strong evidence for bi-directional migration; also referred to as reciprocity. Fig 2 shows the estimates (blue points) and their 95% confidence interval (CI) (blue error bars) for reciprocity from ERGM models that adjust for missing data for each participant. These results imply that reciprocity is a common feature across all 8 LG participants; but its magnitude varies, i.e., there appears to be heterogeneity in the level of reciprocity across participants. For example, the ERGM estimate of reciprocal HIV migration for LG12 is 3.8 (3.2–4.4 95% CI); whereas for both LG04 and LG06 it is 1.2 (1.0–1.4 95% CI). These estimates may be interpreted as the increase (attributable to reciprocity) in the log-odds of a network that contains an edge that results in a unit increase in the number reciprocity pairs compared to a network without that particular edge. We test for heterogeneity using Cochran's Q statistic [20, 21], which yields a value of 46.9 with degrees of freedom of 8, and therefore a p-value of < 0.001–providing strong evidence of heterogeneity in the level of reciprocity. Note that the presence of reciprocity does not imply that the same viral sequence goes back and forth between the tissues, but rather that there is a bi-directional dynamic process of seeding and replenishment.

The smaller estimates for reciprocity from models not adjusting for missing data (shown in red in Fig 2) imply that that missing data has a large impact on reciprocity estimates. Note that adjusting for missing data can lead to either a higher or lower propensity of any particular edge being part of a reciprocal pair. Furthermore, visual inspection of Fig 2, shows that adjusting for missing data increases heterogeneity across hosts. The Cochran's Q statistic without

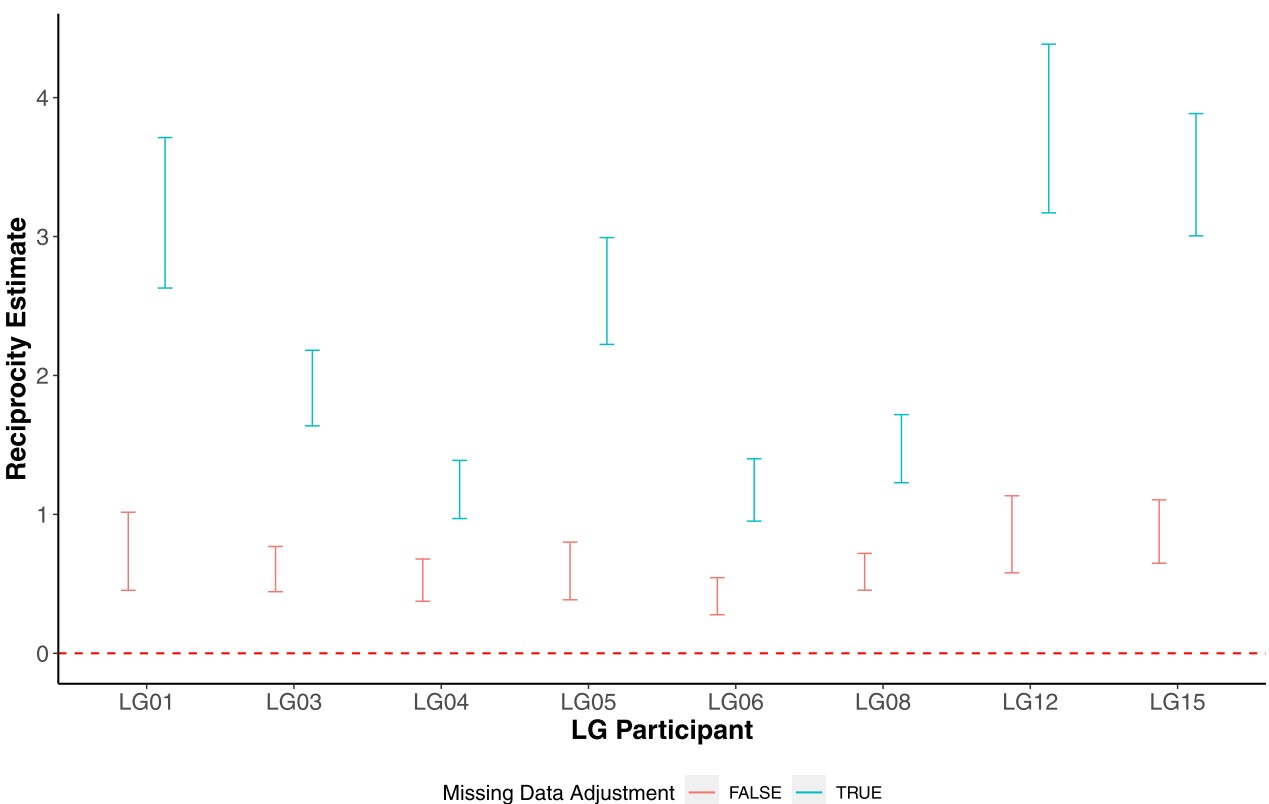

**Fig 2. Estimation of the level of reciprocity in VMNs.** The blue and red points (and error bars) present estimates (and 95% CI) for analyses that did and did not adjust for missing data, respectively.

missing data adjustment is 1.9 (p-value 0.98), which provides no evidence for heterogeneity. Hence, failure to adjust for missing data would lead to an underestimate not only of the level of reciprocity but also its heterogeneity across individuals.

## 2.3 HIV migration events are more common among tissues in the biological system (Homophily)

Another common process that can occur in a network is the tendency of entities (i.e., sampled tissues) to form connections based on similarity of one or more of their individual characteristics. Connection formation is often assortative but can also be dissasortative; the former implies preferential formation of connections between tissues with similar characteristics, and the latter, between tissues with contrasting characteristics [22]. The term homophily is used to describe both process and outcome of preferential connection formation. To assess homophily in this setting, we categorize each tissue based on its biological system. We then assess whether HIV migration events are more common among tissues in the same category. Controlling for the total number of inferred migration events, the propensity of non-zero edges, and the number of HIV sequences obtained from each tissue, our analysis provides strong evidence for presence of homophily in 6 of the 8 participants. Fig 3 shows homophily estimates (blue points) and their 95% confidence interval (CI) (blue error bars) from ERGM models after adjusting for missing data for each participant. For those individuals in whom homophily is detected, there is considerable heterogeneity in its magnitude. For example, LG04 shows little evidence of homophily (log-odds of 0.2, 95% CI 0.1–0.3), whereas LG12 and LG15 show

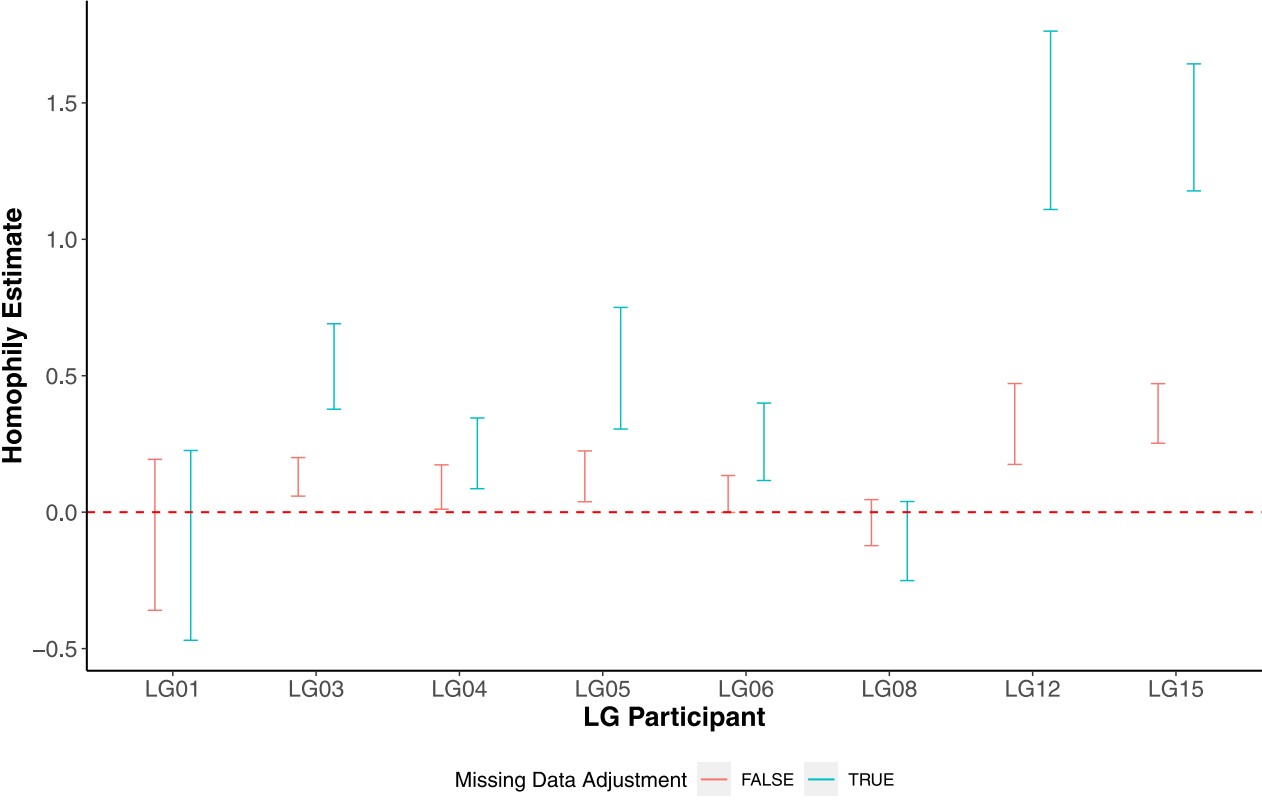

**Fig 3. Estimation of the level of homophily in VMNs.** The blue and red points present estimates for analyses that adjusted and not adjusted for missing data, respectively.

**Table 1. Estimation of the level of homophily in VMNs by tissue type.** Cells with an '*' indicate statistical significance at the 0.05 level.

| System | LG01 | LG03 | LG04 | LG05 | LG06 | LG08 | LG15 |
|---|---|---|---|---|---|---|---|
| Blood | -1.61* | -0.1 | -0.64 | -0.03 | -0.28 | -0.18 | -0.19 |
| CNS | -0.15 | 1.75* | 0.67* | 1.9* | 0.83* | -0.22 | 2.31* |
| Genitourinary tract | 0.15 | -0.11 | -0.03 | -0.03 | 0.02 | -0.44 | -0.09 |
| Gut | 0.09 | 0.35* | 0.22* | -0.86* | 0.47* | 0.12 | 1.98* |
| Lymphoid | -0.01 | 0.29* | 0.17 | 0.52* | 0.03 | -0.21* | -0.06 |
| Other | -0.81 | -0.09 | -0.07 | -0.04 | 0.06 | 0.2 | -0.07 |

significant evidence of homophily (log-odds of 1.4 for both, 95% 1.1–1.8 for LG12 and 1.2–1.6 for LG15). The Cochran's Q statistic provides statistical evidence of heterogeneity of the presence of homophily (p-value of 0.006).

As with reciprocity, adjusting for missing data can have a large impact on the estimates associated with homophily as well as assessing heterogeneity; estimates with no missing data adjustment are shown in red. Fig 3 shows the point estimates and 95% CI for the level of homophily adjusting for missing data (shown in blue) are typically larger than estimates without adjustment (red). In contrast to results adjusted for missing data, the Cochran's Q without adjustment statistic is 2.58 (p-value 0.96), implying no evidence for heterogeneity.

We hypothesize that tissues that are closer in the body have a higher level of homophily. For example, homophily among central nervous system (CNS) tissues can arise from the blood-brain barrier, which has the potential to make the CNS differ from any other organ regarding HIV persistence. Other biological causes of homophily include similarity of cell composition. We estimate a model that can assess the homophily for each tissue type. Controlling for the total number of inferred migration events, the propensity of a non-zero edge, and number of HIV sequences as well as adjusting for missing data, our analysis reveals strong indication of homophily in both CNS tissues and gut in 5 of the 7 participants; the ERGM for participant LG12 did not converge–a known issues with ERGMs [23–25]. Table 1 shows the estimates from the ERGM models for each participant ('*' indicates statistical significance at the 0.05 level). Given the number of tissues sampled for each category, these results should be viewed as preliminary, and we do not present results without adjusting for missing data.

## 2.4 Number of viral migration events is not association with reservoir size (heterogeneity in outward events)

We hypothesize that having large HIV reservoirs (i.e., level of cell-associated HIV DNA) may directly impact the number of viral migration events from tissues. Testing this hypothesis is based on investigation of heterogeneity in outward events among tissues, i.e., assessment of whether tissues with larger HIV reservoirs have a greater or smaller number of outward edges compared to others. Our model controls for the total number of inferred migration events, the propensity of the tissue to have a non-zero number of edges, and the number of HIV sequences. Fitting these models yielded no indication of heterogeneity in outward events based on HIV reservoir size for any of the LG participants (results not shown), i.e., the size of the HIV DNA reservoirs in sampled tissue did not impact the number of outgoing edges.

## 2.5 Viral migration forms a hierarchical structure among the tissues (Triads: Transitivity and 3-cycles)

Transitivity and 3-cycles are network properties that are often considered in the investigation of networks; both involve triads—three node subgraphs. Transitivity is the tendency for

**Table 2. Estimation of the level of triads and 3-cycles in VMNs.** Cells with an '*' indicate statistical significance at the 0.05 level.

| Feature | LG01 | LG03 | LG04 | LG05 | LG06 | LG08 | LG12 | LG15 |
|---|---|---|---|---|---|---|---|---|
| Transitivity | 0.21* | 0.38* | 0.24* | 0.27* | 0.22* | 0.44* | 0.53* | 0.65* |
| Cycles | -0.08 | -0.16* | -0.09* | -0.05 | -0.14* | -0.14* | -0.18 | -0.05 |

migration events to occur between tissues 'A' and 'C' if there are events between tissues 'A' and 'B' and 'B' and 'C'. Cycles consisting of 3-nodes is a feed-back process, but is distinct from reciprocity, which is a 2-node cycle. The results below in Table 2 are based only on the observed data ('*' indicates statistical significance at the 0.05 level); ERGMs adjusting for missing data did not converge. Evidence in support of a greater-than-expected number of network motifs associated with transitivity (as shown by positive and significant estimates), but a less-than-expected number of 3-cycles (negative and significant estimates) implies a hierarchical structure in VMNs.

## 3 Discussion

We investigate five network features: (1) bi-directional flow between tissues; (2) preferential migration among tissues in the same biological system; (3) heterogeneity in level of viral migration with regards to the HIV reservoir size; (4) hierarchical structure of migration; and (5) cyclical migration among several tissues. Due to biological and logistical challenges, not all tissues across a human body are sampled and sequenced; hence, the observations regarding the VMNs are incomplete. We adjust for missing data on host tissues. Our results provide strong evidence for bi-directional flow between tissues, migration among tissues in the same biological system, and hierarchical structure of the viral migration. This analysis provides no evidence that level of viral migration depends on the HIV reservoir size of a tissue. There is evidence in support of the believe that cyclical migration among three tissues occurs less frequently than expected given the amount of viral migration. The analyses also provide evidence for heterogeneity in the presence and magnitude of these features across hosts. Adjustment for missing data has a large impact on our estimates. In particular, adjusting for missing tissue data identifies system-level features within a host as well as heterogeneity in these features across hosts that is not detected without the adjustment.

Evidence for transitivity and against 3-cycles indicates a hierarchical structure in VMNs. This is noteworthy as it tends to imply that few tissues are initial sources of reseeding, thereby further implying that migration propagates through intermediate tissues. This finding complements the absence of evidence for heterogeneity in outward events with respect to HIV reservoir size. Once again, the implication is that VMNs are not characterized by a small number of tissues with large HIV reservoirs serving as the source of direct viral migration to other tissues, that is, our VMNs do not appear to have a hub and spoke network structure with respect to HIV reservoir size. One possible explanation for the absence of heterogeneity in outward events may be that HIV DNA does not reflect HIV replication-competent viruses; our findings might be best confirmed by basing analyses on HIV RNA transcripts measures. Such data are currently being generated for the LG participants. To control for any sample bias (e.g., re-sampling of sequences), our analysis adjusted for the number of unique sequences identified in the tissues. This adjustment may also explain why tissues with a high levels of HIV DNA were not found to be "hubs". [2] found complimentary results–specifically that rebound virus can originate from several cellular and anatomical compartments after treatment interruption [2].

Our investigation has implications for studies aimed at developing HIV curative strategies by identifying features of VMNs, even though it does not reveal the biological mechanisms

driving viral migration. The latter would require deeper study of additional biomarkers—currently underway using the LG cohort. For example, our investigation of homophily shows that tissues within the same biological system are more likely to have migration events among them than tissues that are not within a single system, but the analysis does not reveal the underlying mechanism. Possible explanations for our results are that tissues within a biological system have similar cell composition, a high level of blood exchange, or share other characteristics that promote viral migration. Further analyses using HIV sequences generated at the cellular level are in progress and will be integrated into future models.

Evolution of HIV-1 in a host is shaped by many evolutionary forces, including recombination. If compartmentalization reflects spatial segregation of the virus population, viral recombination is a result of population mixing. Hence, if different point mutations may arise in different tissues, viral migration may bring these variants together and lead eventually to recombination and intermixed viral population. We acknowledge that both migration and recombination should be investigated when studying HIV-1 dynamics within host. While our study does not attempt to evaluate this combined effect and it would require further investigation [26], we investigate the potential impact of intra-host recombination. To do so, we first use GARD to identify potential recombination breakpoints [27]; see S1 Table in S1 File for an overview of the number of putative breakpoint identified. Next, we run our network models (i.e., ERGMs) using the partitioned dataset according to the inferred breakpoint(s). However, these new models exhibit convergence issues; therefore, we cannot provide conclusive assessment of the impact of recombination on features of the viral migration network. Below we provide additional details regarding convergence issues with ERGMs and a alternative network model that does not exhibit such issues, but requires additional methodological development. Furthermore, other factors, such as local immune pressure and antiretroviral therapies, would also need to be considered to comprehensively characterize factors influencing viral dynamics and evolution within host.

Our analyses have several other scientific limitations: it was conducted on 8 participant, some of whom had only a limited number of tissues. Also, there is heterogeneity in the participants, in particular, their terminal disease and ART usage; this heterogeneity may impact the generalizability of our findings. In addition, the construction of the VMNs is based on HIV full-length envelopes sequences (gp160) using single genome dilution techniques. We also include a filtering step to identify of defective or hypermutant sequences [1]. While using HIV envelopes contain less information than full-length genome, they are less likely to be impacted by amplification failure of long fragment. Furthermore, full-length genome may miss a large proportion of intact proviruses due to amplification failure [28]. Our findings also can be impacted by blood T cell contamination of tissue samples obtained during autopsy. Previous sequence analyses on the samples showed viral compartmentalization for all participants, which suggests that possible blood contamination would not negate our findings; see Chaillon et al. [1] for additional details regarding contamination. With regard to statistical issues, ERGMs can suffer from degeneracy as seen in some of the analyses, including transitivity and 3-cycles [23–25]. Another limitation of ERGMs is that the theoretical foundation for estimation of standard error associated with the ERGMs has not been fully developed–even for completely observed networks [29]. Additional development is needed for to incorporate the additional uncertainty that arises from missing data in estimates of confidence intervals. Such uncertainty can be captured using a Bayesian paradigm. Bayesian extensions of the ERGMs– referred to as BERGMs–has been developed, which can capture such uncertainty in estimating credible intervals [8]. These approaches, however, do not currently allow for analyzing valued networks. In addition, there are computational limits to the size of networks that can be analyzed using BERGMs. A potential future direction is investigating complex features using the

congruence class model (CCM) for networks [30–34]. CCMs form a broad class that includes as special cases such common network models as the Erdős-Rényi-Gilbert and stochastic block models as well as many ERGMs. CCMs requires additional methodological development to address missing network data, which is necessary for our context of missing tissues; such methodological development is currently underway.

The primary goal of our results and investigation is to understand the potential (and necessity) of analyzing viral migration using network science techniques. While phylodynamic modeling has greatly enhanced these efforts, our analysis provides additional insights through analysis of migration as a network—rather than as pairwise events between two tissues. Insights gained using network science techniques in analysis of VMNs have therapeutic implications, in that they may aid in the identification of common features in viral migration in people with HIV (or a subpopulation, such as those who interrupt ART). An understanding of these features may elucidate potential processes to target the source of viral reseeding. For example, our findings suggest a hierarchical structure for viral migration among the tissues. Treatments targeting tissues upstream may be more efficient in preventing viral rebound compared to treatments focused on tissues further down in the viral migration structure. Therefore, this research may serve as initial insight into developing more efficient treatments to provide viral migrating and reseeding of tissues.

## 4 Methods

### 4.1 Last Gift Cohort

The data derives from the LG Cohort, which is a cohort from an end-of-life HIV research study underway at the University of California San Diego. The goal of the study is to investigate HIV reservoirs using a rapid autopsy procedure in PWH who voluntarily agree to have their organs harvested post-mortem [1, 14]. To accomplish this goal, the study collects detailed clinical data and biological samples from participants before death and then collects additional samples during a rapid autopsy procedure [35]. Table 3 presents the demographic characteristics of the participants and summary statistics of their VMN, including the number of observed and missing tissues and the number of directed edges (i.e., the presence of a migration event inferred from Bayesian models). S2 Table in S1 File provides information on the LG participant number, tissue name, system category, and number of HIV sequences from each tissue sample. All HIV sequences—except for blood plasma samples—are proviral DNA sequences. The genomic DNA is extracted, and precipitation is performed to concentrate DNA. Concentrations of DNA are determined using NanoDrop One (ThermoScientific). We perform single genome dilution and sequencing of full-length envelope (gp160). See the S1

**Table 3. Demographic characteristics of the participants and summary statistics of their VMN, including the number of observed and missing tissues and the number of directed edges (i.e., the presence of a migration event inferred from Bayesian models).**

| ID | Gender | On ART at death | Cancer Cancer | Observed tissues | Missing tissues | Observed migration |
|------|--------|-----------------|---------------|------------------|-----------------|--------------------|
| LG01 | M | Yes | No | 10 | 23 | 23 |
| LG03 | M | No | Yes | 19 | 14 | 87 |
| LG04 | M | Yes | No | 25 | 8 | 151 |
| LG05 | M | No | No | 13 | 20 | 36 |
| LG06 | M | No | Yes | 20 | 13 | 100 |
| LG08 | M | No | Yes | 20 | 13 | 122 |
| LG12 | F | No | Yes | 7 | 26 | 20 |
| LG15 | M | Yes | No | 11 | 22 | 41 |

File in [1] for additional information on collection and processing of data from the LG Cohort.

## 4.2 Bayesian discrete phylogeographic models

In this study, we use inferred migration events between tissues obtained from Bayesian discrete phylogeographic models using HIV full length envelopes sequences sampled in each tissue [36, 37]. Briefly, discrete trait analyses (DTA) consider spatial diffusion among discrete locations, from which viral sequences have been sampled, as a continuous-time Markov process [36]. From a statistical perspective, this Bayesian stochastic search variable sampling (BSSVS) procedure is particularly appropriate because statistical inference is efficient (achieves lowest variance). Importantly, it also uses a Bayes Factor (BF) test to infer the most parsimonious description of the diffusion process [36]. With the BSSVS procedure, BF support for all possible migration rates are obtained in a single DTA analysis [36]. As this procedure only accounts for the number of trait states (e.g. locations), it remains difficult to assess significant support for a particular migration link. To remedy this problem [36], developed a new measure of significance (the adjusted BF test) that has a low false-positive rate by incorporating information on the relative abundances of samples from each location in the data set. The new measure of support for particular migration relies on the a priori expected and a posteriori noted inclusion frequencies under BSSVS. See Chaillon et al. [1] for additional details.

## 4.3 Exponential random graph models

We use ERGMs to make inference regarding features in the presence of missing data. Let $Y$ be the space of all potential directed valued networks with $N$ tissues. In our setting $N = 33$, the number of unique tissues sampled across all LG participants. The values represent counts of migration event among the tissues. Let $y \in Y$ and let $y_{i,j}$ be the value for edge $(i, j)$ where $i$ and $j$ denote tissues. The probability mass function (PMF) defined by an ERGM on $Y$ is the following PMF:

$$P_{h,g}(Y = y; \theta) = \frac{h(y)exp(\theta^T g(y))}{\kappa_{h,g}(\theta)}, \tag{1}$$

where $\theta$ is our vector of parameters, $g(y)$ is a vector of summary statistics of $y$, $h$ specifies a reference measure, and $\kappa_{h,g}(\theta)$ is a normalizing constant; see Krivitsky et al. [38] for additional details.

Our analyses investigate network models associated with each of our features. Each of the models control for the total amount of migration, the propensity of a non-zero edges, and number of HIV sequences. We define a tissue as missing if the tissue sample is not available for an individual but is collected for at least one other participant in the LG cohort. To account for missing data, we assume that the pattern of missingness is ignorable; that is, the probability of a value being missing only depends on the observed data [7]. The likelihood for model parameters is then calculated by marginalizing the ERGM likelihood over all possible complete networks that are compatible with the observed data. The models are estimated using the STATNET package [39, 40] in R CRAN [41].

## 4.4 Assessing heterogeneity

To investigate heterogeneity in the effects of mechanisms across participants, we use both graphical and statistical assessments. For our graphical assessment, we investigate whether confidence intervals for the parameter associated with a particular mechanism across

individual have little overlap. Such a pattern generally indicates the presence of heterogeneity. To formally test for heterogeneity, we use the Cochran's Q statistic [20], shown below:

$$Q = n \sum_{i=1}^{n} w_i * (\hat{\theta}_i - \hat{\theta})^2, \tag{2}$$

where $\hat{\theta}$ is the mean estimate across all LG participants for the network property of interest, $\hat{\theta}_i$ is the estimate for LG participant $i$, and $w_i$ is the inverse of the variance for participant $i$. The sum is across all $n = 8$ LG participants. Due to weights on the items in the sum, the value of $Q$ depends not only on the deviation of $\hat{\theta}_i$ from $\hat{\theta}$, but also on the precision of participant estimates. The Cochran's Q statistic approximately follows a chi-squared distribution, with $n - 1$ degrees of freedom.

## Supporting information

**S1 File. Contains all the supplementary tables.**
(PDF)

## Author Contributions

**Conceptualization:** Ravi Goyal, Victor De Gruttola, Davey M. Smith, Antoine Chaillon.

**Data curation:** Ravi Goyal, Sara Gianella, Gemma Caballero, Magali Porrachia, Caroline Ignacio, Brendon Woodworth, Davey M. Smith, Antoine Chaillon.

**Formal analysis:** Ravi Goyal, Antoine Chaillon.

**Funding acquisition:** Davey M. Smith.

**Investigation:** Ravi Goyal, Sara Gianella, Davey M. Smith, Antoine Chaillon.

**Methodology:** Ravi Goyal, Victor De Gruttola, Davey M. Smith, Antoine Chaillon.

**Software:** Ravi Goyal.

**Supervision:** Victor De Gruttola, Davey M. Smith, Antoine Chaillon.

**Validation:** Ravi Goyal, Sara Gianella.

**Visualization:** Antoine Chaillon.

**Writing – original draft:** Ravi Goyal, Victor De Gruttola, Antoine Chaillon.

**Writing – review & editing:** Ravi Goyal, Victor De Gruttola, Sara Gianella, Gemma Caballero, Magali Porrachia, Caroline Ignacio, Brendon Woodworth, Davey M. Smith, Antoine Chaillon.

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
