## [Decision Letter · Decision Letter 0]

7 Feb 2023

PONE-D-22-24111Identification of system-level features in HIV migration within a hostPLOS ONE

Dear Dr. Goyal,

Thank you for submitting your manuscript to PLOS ONE. After careful consideration, we feel that it has merit but does not fully meet PLOS ONE’s publication criteria as it currently stands. Therefore, we invite you to submit a revised version of the manuscript that addresses the points raised during the review process.

Dear Authors

Based on the comments from the reviewers, the manuscript needs a minor revision before acceptance. There are a few concerns by reviewers. Please address those.

We look forward to receiving your revised manuscript.

Kind regards,

Nafees Ahemad

Academic Editor

PLOS ONE

Journal Requirements:

2. In the Methods section of your revised manuscript, please include the full name of the institutional review board or ethics committee that approved the protocol, the approval or permit number that was issued, and the date that approval was granted.

Please note that PLOS ONE has specific guidelines on code sharing for submissions in which author-generated code underpins the findings in the manuscript. In these cases, all author-generated code must be made available without restrictions upon publication of the work. Please review our guidelines at https://journals.plos.org/plosone/s/materials-and-software-sharing#loc-sharing-code and ensure that your code is shared in a way that follows best practice and facilitates reproducibility and reuse.

“This research is supported by grants from the National Institutes of Health (P01 AI131385-05, R01 AI-147441, DA051915, R01 DK131532, P01 AI169609)

Sponsors did not play any role in the manuscript.”

“None”

6. Please update your submission to use the PLOS LaTeX template. The template and more information on our requirements for LaTeX submissions can be found at http://journals.plos.org/plosone/s/latex.

7. Thank you for stating the following in the Acknowledgments Section of your manuscript:

“This research is supported by grants from the National Institutes of Health (P01 AI131385-05, R01 AI-147441, DA051915, R01 DK131532, P01 AI169609). Conflict of Interest: None.”

“This research is supported by grants from the National Institutes of Health (P01 AI131385-05, R01 AI-147441, DA051915, R01 DK131532, P01 AI169609

Sponsors did not play any role in the manuscript.”

Additional Editor Comments (if provided):

Dear Authors

Based on the comments from the reviewers, the manuscript needs a minor revision before acceptance. There are a few concerns by reviewers. Please address those.

Reviewers' comments:

Reviewer's Responses to Questions

**Comments to the Author**

1. Is the manuscript technically sound, and do the data support the conclusions?

Reviewer #1: Yes

Reviewer #2: Partly

2. Has the statistical analysis been performed appropriately and rigorously? 

Reviewer #1: Yes

Reviewer #2: I Don't Know

3. Have the authors made all data underlying the findings in their manuscript fully available?

Reviewer #1: Yes

Reviewer #2: No

4. Is the manuscript presented in an intelligible fashion and written in standard English?

Reviewer #1: Yes

Reviewer #2: Yes

5. Review Comments to the Author

Reviewer #1: Goyal and colleagues utilize unique and diverse tissue samples from the Last Gift cohort to assess HIV DNA sequencing data to study the “migration” of HIV sequences across and between tissues. This is an interesting methodological and data analysis approach to questions that are difficult to ask, especially as access to tissues is limited. The approaches have several technical limitations, which may currently be unavoidable, that should be acknowledged and discussed.

As has been recently discussed (White JA et al. PLoS Pathog. 2022. PMID: 36074794) near-full-length PCR techniques may introduce bias and over-represent some sequences. How could such bias influence the representation of sequences and their apparent "flow" or change in proportion over time? Further most sequences are defective, but may proliferate as their host cells proliferate, Both myeloid and lymphoid cells may carry HIV DNA, and some cells may migrate within tissue to various body compartments, without true migration of viral particles due to spreading infection. These biological aspects should be discussed in the methodological description of the analysis and its interpretation.

Reviewer #2: 1) The paper states that data and software will be available from the authors upon request, but the proper way to make the data available is to create GenBank entries for the HIV sequences and list the accession numbers in the publication. It is also very nice of multiple sequence alignments, or other useful data formats are stored at TreeBase, or the data DRYAD or similar online repositories.

2) Intra-patient viral recombination is not mentioned in the paper. It is difficult to analyze intra-subtype recombination in HIV-1, and even more difficult to analyze intra-patient recombination. However, tools such as the Highlighter too at the LANL HIV Database ( https://www.hiv.lanl.gov/content/sequence/HIGHLIGHT/highlighter_top.html ) and GARD ( http://www.datamonkey.org/GARD/ ) can be useful in identifying whether or not recombination is likely to be influencing phylogenetic analyses.

3)The Figure 1 legend does not mention it, but I assume the overall size or diameter of each patient graph is proportional to virus diversity in that patient. So for example LG12 fig1G had less diverse virus than LG01 fig1A.

4) There are many sentences which don't make sense to me, and I wonder if it is because words are missing? For example on page 15 "While phylodynamic modeling has

greatly enhanced these e orts, our analysis provides additional insights through analysis of migra-

tion as a network|rather than as pairwise events tissue." Maybe was supposed to end with "pairwise events between two tissues."?

5) I think the paper could benefit from a better description of how this type of study can help with a cure. The paper says "Insights gained using network science techniques in analysis of VMNs have therapeutic implications, in that they may aid in the identification of common features in viral migration, and, by facilitating the targeting of specific processes, potentially increase efficacy of HIV cure." Three of the 8 patients were not on ART at the time of death, and might have therefor had higher viral loads than the others.

6) The paper has quite a bit of discussion of the computational analyses, but no information is provided about the data acquisition. How were the tissues sampled to reduce or eliminate the potential for sampling blood cells rather than tissue cells in each tissue? Are the sequences likely to be from viral RNA or from proviral DNA integrated into the host genome? Was the complete envelope gp160 region sequenced? What were the 33 tissues for each patient? Table 3 shows 7 tissues sampled and 26 missing for patient LG12 and Gig 1G shows 7 nodes with 5 of them being from gut. Most patients seem to have just one or two blood tissues sampled. In many places, the paper says "number of HIV sequences" but nowhere is it mentioned whether there ere hundreds of sequences from each tissue sample, or dozens, or thousands.

6. PLOS authors have the option to publish the peer review history of their article (what does this mean?). If published, this will include your full peer review and any attached files.

Reviewer #1: No

Reviewer #2: **Yes: **Brian T. Foley

---

## [Editor Report · Decision Letter 1]

29 Aug 2023

Identification of system-level features in HIV migration within a host

PONE-D-22-24111R1

Dear Dr. Goyal,

We’re pleased to inform you that your manuscript has been judged scientifically suitable for publication and will be formally accepted for publication once it meets all outstanding technical requirements.

Kind regards,

Nafees Ahemad

Academic Editor

PLOS ONE
---

## [Editor Report · Acceptance letter]

11 Sep 2023

PONE-D-22-24111R1 

Identification of system-level features in HIV migration within a host 

Dear Dr. Goyal:

I'm pleased to inform you that your manuscript has been deemed suitable for publication in PLOS ONE. Congratulations! Your manuscript is now with our production department. 

Kind regards, 

on behalf of

Dr. Nafees Ahemad 

Academic Editor

PLOS ONE